# 4-*O*-Glucosylation of Trichothecenes by *Fusarium* Species: A Phase II Xenobiotic Metabolism for t-Type Trichothecene Producers

**DOI:** 10.3390/ijms222413542

**Published:** 2021-12-17

**Authors:** Kosuke Matsui, Hirone Takeda, Koki Shinkai, Takao Kakinuma, Yoshiaki Koizumi, Masahiro Kase, Tomoya Yoshinari, Hiroaki Minegishi, Yuichi Nakajima, Shunichi Aikawa, Naoko Takahashi-Ando, Makoto Kimura

**Affiliations:** 1Graduate School of Bioagricultural Sciences, Nagoya University, Furo-cho, Chikusa-ku, Nagoya 464-8601, Aichi, Japan; matsuik.toyo@gmail.com (K.M.); nakajima.yuichi@a.mbox.nagoya-u.ac.jp (Y.N.); mkimura@agr.nagoya-u.ac.jp (M.K.); 2Graduate School of Science and Engineering, Toyo University, 2100 Kujirai, Kawagoe 350-8585, Saitama, Japan; s36d02000102@toyo.jp (H.T.); s16d01301215@gmail.com (K.S.); s36d02000045@toyo.jp (Y.K.); s36D02100079@toyo.jp (M.K.); minehiro@toyo.jp (H.M.); 3Faculty of Science and Engineering, Toyo University, 2100 Kujirai, Kawagoe 350-8585, Saitama, Japan; tt19970815@gmail.com; 4Division of Microbiology, National Institute of Health and Sciences, Kawasaki 210-9501, Kanagawa, Japan; yoshinari@nihs.go.jp; 5Research Institute of Industrial Technology, Toyo University, 2100 Kujirai, Kawagoe 350-8585, Saitama, Japan; s-aikawa@toyo.jp

**Keywords:** d-type trichothecene, *Fusarium graminearum*, glucosylation, t-type trichothecene, *Tri101* resistance gene, phase II xenobiotic metabolism

## Abstract

The t-type trichothecene producers *Fusarium sporotrichioides* and *Fusarium graminearum* protect themselves against their own mycotoxins by acetylating the C-3 hydroxy group with Tri101p acetylase. To understand the mechanism by which they deal with exogenously added d-type trichothecenes, the Δ*tri5* mutants expressing all but the first trichothecene pathway enzymes were fed with trichodermol (TDmol), trichothecolone (TCC), 8-deoxytrichothecin, and trichothecin. LC-MS/MS and NMR analyses showed that these C-3 unoxygenated trichothecenes were conjugated with glucose at C-4 by α-glucosidic linkage. As t-type trichothecenes are readily incorporated into the biosynthetic pathway following the C-3 acetylation, the mycotoxins were fed to the Δ*Fgtri5*Δ*Fgtri101* mutant to examine their fate. LC-MS/MS and NMR analyses demonstrated that the mutant conjugated glucose at C-4 of HT-2 toxin (HT-2) by α-glucosidic linkage, while the Δ*Fgtri5* mutant metabolized HT-2 to 3-acetyl HT-2 toxin and T-2 toxin. The 4-*O*-glucosylation of exogenously added t-type trichothecenes appears to be a general response of the Δ*Fgtri5*Δ*Fgtri101* mutant, as nivalenol and its acetylated derivatives appeared to be conjugated with hexose to some extent. The toxicities of 4-*O*-glucosides of TDmol, TCC, and HT-2 were much weaker than their corresponding aglycons, suggesting that 4-*O*-glucosylation serves as a phase II xenobiotic metabolism for t-type trichothecene producers.

## 1. Introduction

Trichothecenes are a group of mycotoxins that contain a trichothecene skeleton (12,13-epoxy-trichothec-9-ene; EPT) in their structures (Figure 1). To date, many trichothecenes with various modifying groups have been found. They are produced by several genera of fungi, including *Fusarium*, *Spicellum*, *Trichothecium*, *Trichoderma*, and *Myrothecium*. Based on their chemical structures, Ueno classified trichothecenes into four types: A to D [1]. Type A trichothecenes are defined as those lacking a ketone at C-8, including T-2 toxin, HT-2 toxin (HT-2), 4,15-diacetoxyscirpenol (DAS), neosolaniol (NEO), 8-deoxytrichothecin (8-deTCN), and harzianum A. The former four toxins are produced by *Fusarium*, such as *F. sporotrichioides* and *F. langsethiae*, and the latter two toxins are produced by non-fusaria species. On the other hand, type B trichothecenes, including deoxynivalenol (DON), nivalenol (NIV), trichothecolone (TCC), and trichothecin (TCN), possess a ketone at C-8. While wheat scab fungus, such as *F. graminearum* and *F. culmorum*, contaminates grains with DON- and NIV-type trichothecenes, *Trichothecium* and other genera produce TCC and TCN. Type C trichothecenes have a second epoxide ring between the C-7 and C-8 positions, and type D trichothecenes are macrocyclic trichothecenes with polyketide esters between the C-4 and C-15 positions [2].

There is also a biological classification of trichothecenes proposed by Kimura et al., by which they are divided into d- and t-type on the basis of the importance of the C-3 position in the self-protection of the trichothecene producers [3,4]. In the biosynthesis of d-type trichothecenes, a bicyclic intermediate, isotrichodiol, is cyclized to give the first trichothecene intermediate EPT that lacks a C-3 hydroxy group [5]. In contrast, in the biosynthesis of t-type trichothecenes, such as DON, additional oxygenation yields the precursor, isotrichotriol [6]. This bicyclic intermediate is cyclized to isotrichodermol (ITDmol), the first tricyclic intermediate with an EPT skeleton and a C-3 hydroxy group [7,8]. While t-type trichothecenes are mainly produced by fusaria, C-3 unoxygenated d-type trichothecenes are produced by non-fusaria species.

The idea of the d- and t-type classification is based on the result of the feeding experiment of Zamir and colleagues [9]. They reported that the 3-*O*-acetyl group of the tricyclic intermediate isotrichodermin (ITD), which is mostly deacetylated during feeding to the *Fusarium* culture, must be re-acetylated for the DON biosynthetic pathway to proceed. Thus, the 3-*O*-acetylation of ITDmol to ITD was hypothesized to work as a self-protection mechanism through the metabolic shielding of trichothecenes [3]. Indeed, 3-acetyltrichothecenes are generally less toxic than their corresponding 3-*O*-hydroxyl cognates [3,10,11]. The relevant *Tri101* resistance gene responsible for the 3-*O*-acetylation was isolated from *F. graminearum* [3] and *F. sporotrichioides* [12]. Independently, McCormick et al. cloned *Tri101* from *F. sporotrichioides* [7]. With a targeted gene disruption experiment, they showed that the C-3 acetylase gene was essential for modifications of the trichothecene skeleton. The transgenic expression of *Tri101* in yeasts [3,7], tobacco [13], rice [14], and cultured mammalian cells [15] conferred a significant level of resistance to trichothecenes.

A modifying group at the C-3 position of trichothecenes has a significant impact on their toxicity. Shima et al. isolated a soil bacterium that ketonizes DON at C-3 by utilizing an enrichment culture and demonstrated a remarkable decrease in its immunosuppressive toxicity [16]. He et al. isolated an aerobic bacterium capable of converting DON to 3-*epi*-DON [17]; they unambiguously demonstrated the abrogation of DON toxicity by the epimerization of the C-3 hydroxyl group [18]. The responsible detoxifying enzymes, DepA [19] and DepB [20], and their coding genes were identified in *Devosia mutans* and the enzymatic epimerization proved to proceed through the formation of the 3-*keto*-DON intermediate [21].

It has long been known that wheat has in vitro detoxifying activity toward DON [22]. Sewald et al. first isolated the plant metabolite of DON from the maize suspension culture and elucidated its structure as DON-3-*O*-β-glucopyranoside (DON-3-glc) by NMR [23]. Later, a UDP-glucosyltransferase (UGT) gene responsible for DON glucosylation was cloned from *Arabidopsis*
*thaliana*, and the plant mechanism of trichothecene detoxification was confirmed [24]. Subsequently, UGTs have been identified in wheat, barley, *Brachypodium*, and rice [25,26,27]. Although the glucoside linkage of such 3-*O*-glucoconjugates is relatively stable, the aglycones are released by microbial hydrolysis in the intestines of animals and could recover their toxicity [28,29,30]. As DON-3-glc evades detection by a conventional DON analysis method, their potential toxicity has become a growing concern for food safety and public health [31].

As described above, there is abundant evidence that the C-3 modification of t-type trichothecenes plays a critical role in the attenuation of their toxicity. However, such resistance mechanisms are not possible against d-type trichothecenes due to the lack of a hydroxy group at C-3. According to Proctor et al., the biosynthetic genes of the d- and t-type trichothecene-producing fungi are evolutionarily quite different [32]. Therefore, to gain insight into the different mechanisms, we characterized the biosynthetic enzymes by feeding the d- and t-type trichothecenes to various trichothecene-producing strains in different fungal genera. When the d-type trichothecene trichodermol (TDmol) was fed to *F. sporotrichioides* and *F. graminearum*, it was not incorporated into the biosynthetic pathway of the t-type trichothecenes. Instead, hexose conjugation at C-4 seemed to have occurred [33]. In this study, we analyzed the hexose conjugate product of TDmol by NMR. We also examined whether hexose conjugation occurs at C-4 of the t-type trichothecenes when fed to the *Tri101* disruptant of *F. graminearum*, which has no means of protection against mycotoxins added exogenously.

## 2. Results and Discussion

### 2.1. Feeding of d-Type Trichothecenes to t-Type Trichothecene Producers

The *Tri5* deletion mutants of fusaria do not produce trichothecenes but they still express other trichothecene enzymes. Thus, they can be used as convenient tools for feeding experiments [34,35,36]. Here, the d-type trichothecenes, TDmol, 8-deTCN, TCC, and TCN, were fed to the *F. sporotrichioides* strain FsΔ*Tri5* (Δ*Fstri5::hph*), and ethyl acetate extracts of the cell culture at selected time points were subjected to TLC (Figure 2a,b). Interestingly, the spots of these d-type trichothecenes disappeared by 48 h at the latest, suggesting that the mycotoxins were modified to metabolites with higher hydrophilicity. Thus, the culture medium was extracted with acetonitrile and the extract was applied to LC-MS/MS. The theoretical *m*/*z* values of the ammonium adduct ion of TDmol, 8-deTCN, TCC, and TCN were 268.1907, 336.2169, 282.1700, and 350.1962, respectively. In agreement with the TLC results, all the corresponding peaks of the extracted ion chromatograms (XIC) disappeared between 6 and 24 h. In exchange for these peaks, new peaks of 430.244 ± 0.025 emerged around 2.84 min in the extracts fed with TDmol and 8-deTCN, the MS/MS spectra of which coincided with those obtained by TDmol feeding in the previous report [33]. Similarly, new peaks of *m*/*z* 444.223 ± 0.025 emerged around 2.37 min in the extracts fed with TCC and TCN. 

The fate of these non-precursor trichothecenes fed to *F. sporotrichioides* sharply contrasts with that of ITDmol, as this t-type trichothecene precursor is readily metabolized to T-2 toxin [8]. Indeed, the strain FsΔ*Tri5* completely metabolized ITDmol to T-2 toxin. We also fed these four d-type trichothecenes to the *F. graminearum* strain FGD5 (Δ*Fgtri5::neo*) [6] and observed the same shift in the XIC peaks. In the feeding with FGD5, ITDmol was metabolized to 4,15-diacetylnivalenol (4,15-diANIV) and 4-acetylnivalenol (4-ANIV), the final products of *F. graminearum* MAFF 111233.

According to the previous study [33], the XIC peak at 2.84 min of *m*/*z* 430.2435 obtained by the TDmol feeding (corresponding to 2.99 min in the literature) was assumed to be [TDmol-4-*O*-hexoside (C_6_H_10_O_5_) + NH_4_]^+^, as the C-4 hydroxy group is the only possible substituent to be conjugated with hexose. The same 2.84 min peak in XIC obtained by the 8-deTCN feeding in the present study was, thus, speculated to be TDmol-4-*O*-hexoside (**1**), as the MS/MS spectra were perfectly superimposable (Appendix A). Similarly, the XIC peaks of the theoretical *m*/*z* 444.2228 around 2.37 min in the TCC and TCN feedings showed the same MS/MS spectra (Figure 2c; upper and middle panels). The spectra were similar to those of [TCC (C_15_H_20_O_4_) + NH_4_]^+^ (theoretical *m*/*z* 282.1700) below an *m*/*z* value of 265.1437 (Figure 2c; lower panel). In analogy to the results of the TDmol and 8-deTCN feedings, we speculated that these metabolites were TCC-4-*O*-hexoside (**2**).

Given that these non-precursor d-type trichothecenes are ultimately conjugated with hexose at C-4 by the t-type trichothecene producers, the butenoyl group must first be removed for subsequent hexose conjugation as follows: 8-deTCN→TDmol→TDmol-hexoside, and TCN→TCC→TCC-hexoside. The time course of disappearance of these C-4 acylated d-type trichothecenes supports this hypothesis (Figure 2a,b).

### 2.2. Purification and Structure Elucidation of Hexose Conjugate of d-Type Trichothecenes

To confirm the structures of **1** and **2**, we carried out large-scale preparations of these putative trichothecene glucoconjugates. In total, 80 mg of TDmol was fed to 2.0 L of GYEP medium inoculated with FsΔ*Tri5*. After extraction with acetonitrile, the metabolite (16.2 mg) was fractionated as described in Section 3.7. Briefly, the metabolites were applied to Purif^®^-Rp2 equipped with a reverse phase column; the partially purified **1** (10.2 mg) was then purified by preparative HPLC twice. The final yield of **1** was 6.0 mg from 40 mg of TDmol fed to FsΔ*Tri5*. NMR analysis identified **1** as TDmol-4-*O*-α-glucopyranoside (TDmol-4-glc) (Figure 3 and Appendix A, Appendix A). We also fed 72 mg of TCC to the strain FsΔ*Tri5* inoculated in 1.8 L of GYEP medium to purify **2** in the same manner; the yield was 12 mg after acetonitrile extraction, 8 mg after Purif-Rp2 purification, and 4.8 mg after HPLC. NMR analysis identified **2** as TCC-4-*O*-α-glucopyranoside (TCC-4-glc) (Figure 3 and Appendix A, Appendix A). These results clearly demonstrate that 4-oxygenated d-type trichothecenes are conjugated with glucose at C-4 when fed to the culture of t-type trichothecene producers.

Interestingly, Gorst-Allman et al. have reported on the structure elucidation of 15-acetoxyscirpenol-4-*O*-α-glucopyranoside (15-AS-4-glc) purified from the maize meal culture (5 kg) of *Fusarium sulphureum* [37] more than 35 years ago. The metabolite was identified as giving a blue color reaction with 4-(*p*-nitrobenzyl)pyridine (NBP) on a TLC plate. The yield of 15-AS-4-glc was 182 mg under the condition when DAS, in the same culture, was recovered as much as 3.02 g [38]. This suggests that the C-4 glucosylation participates in a minor shunt pathway in the biosynthesis of DAS under certain culture conditions. To the best of our knowledge, this is the only report on the structural confirmation of 4-*O*-glucoside of a t-type trichothecene by NMR. The observation that *Fusarium* species show such 4-*O*-glucosylation activity, regardless of the type of 4-hydroxytrichothecenes, may indicate conserved features of t-type trichothecene producers to the exposure of C-4 oxygenated trichothecenes.

### 2.3. Glucosylation of a t-Type Type A Trichothecene, HT-2, by a Tri101-Null Mutant of F. graminearum

Except for the case of the *F. sulphureum* large-scale culture [37], there have been no reports so far on the production of t-type trichothecene 4-*O*-glucosides in axenic cultures of *Fusarium* strains. Considering the limited accumulation of 15-AS-4-glc on the maize meal medium, it seems likely that the amounts of 4-*O*-glucosylated trichothecenes are marginal in a small-scale liquid culture. It is also possible that a C-3 hydroxy group, as well as C-4, must not be acylated for efficient conjugation with glucose, as the occurrence of 3,15-diacetoxyscirpenol-4-*O*-glucoside was not reported in the literature [37].

Since the presence of a C-3 acetyl group is essential to serve as the substrates of the trichothecene pathway enzymes, including the C-4 acetylase Tri7p [39], the lack of the 3-*O*-acetyl group makes it easier for non-trichothecene enzymes to act on the C-4 hydroxy group. For this reason, the glucose conjugate of trichothecenes could be made more easily by *Fusarium* when added exogenously to a *Tri101* disruption mutant. To assess this possibility, we examined the metabolic fate of HT-2, a t-type type A trichothecene hydroxylated at both C-3 and C-4, in the feeding experiment. *F. graminearum* FGD5 and FGD5/101 (Δ*Fgtri101::hph*, Δ*Fgtri5::neo*) mutants, with or without trichothecene 3-*O*-acetyltransferase activity, respectively, were used. When fed to the FGD5 mutant, HT-2 was metabolized to 3-acetyl HT-2 toxin (3-A HT-2) and T-2 toxin, as revealed by TLC and LC-MS/MS analyses (Figure 4a). In contrast, HT-2 fed to FGD5/101 did not yield the same metabolites on TLC. LC–MS/MS analysis of the feeding mixture revealed the emergence of a new XIC peak at 24 h (Figure 4b), which corresponded to [HT-2-hexoside (C_28_H_42_O_13_) + NH_4_]^+^ (theoretical *m*/*z* 604.2964). The MS/MS spectra of the parent ion *m*/*z* 604.2964 including its fragment *m*/*z* 425.2169 (Figure 4c) strongly suggested that HT-2 was transformed into the corresponding hexose conjugate **3**, as the spectra were quite similar to those of [HT-2 (C_22_H_32_O_8_) + NH_4_]^+^ (theoretical *m*/*z* 442.2435) below an *m*/*z* value of 425.2163 (Figure 4d). Thus, HT-2 added exogenously was suggested to be metabolized along pathways differently between the FGD5 and FGD5/101 mutants (Figure 4a,b). To confirm the structure of **3**, a total of 300 mg HT-2 was fed to FGD5/101 in 15 L of the rice flour (RF) medium, and 35 mg of acetonitrile extract was obtained. After fractionation by Purif^®^-Rp2 (yield 24 mg), followed by repeated purification by preparative HPLC, 12 mg of pure **3** was obtained. NMR analysis confirmed that the structure of **3** was HT-2 toxin-4-*O*-α-glucopyranoside (HT-2-4-glc) (Figure 5 and Appendix A, Appendix A).

To examine whether 3-acetyltrichothecenes can serve as the substrates of *Fusarium* UGT(s), 3-A HT-2 was fed to the strains FGD5 and FGD5/101. The XIC peak corresponding to the HT-2-4-glc ammonium adduct ion was detected from the FGD5/101 culture, which could reasonably be explained by the deacetylation of 3-A HT-2 and the subsequent C-4 glucosylation of HT-2. In contrast, 3-acetyl HT-2 toxin-4-α-glucopyranoside was not found in the feeding extract of either mutant strain. We also incubated HT-2 and **3** individually with Tri101p and acetyl CoA to determine whether C-3 can be acetylated when C-4 is glucosylated. Although more than 88% of HT-2 was converted to 3-A HT-2 in 1 h, acetylated **3** could not be detected under the same condition. Only when a high concentration of Tri101p was used and the reaction time was extended to 6 h, a small amount of **3** acetylated at C-3, with a parent ion peak of *m*/*z* 646.3076 corresponding to [3-A HT-2-hexoside (C_30_H_44_O_14_) + NH_4_]^+^ (theoretical *m*/*z* 646.3069), was barely detectable in MS/MS analysis (Appendix A; upper panel). The MS/MS spectra of the putative 3-A HT-2 glucoside below an *m*/*z* value of 484.2635 was similar to that of 3-A HT-2 (theoretical *m*/*z* 484.2541) (Appendix A; lower panel). In this way, **3** proved to be much less efficiently acetylated by Tri101p than its aglycon HT-2. The two modifications, 3-*O*-acetylation and 4-*O*-glucosylation, marginally occur simultaneously on the t-type type A trichothecene ring.

### 2.4. Glucosylation of t-Type Type B Trichothecenes by a Tri101-Null Mutant of F. graminearum

To determine whether fusaria can also glucosylate t-type type B trichothecenes oxygenated at C-4, FGD5 and FGD5/101 were fed with NIV and 15-acetylnivalenol (15-ANIV). The mutant strains were also fed with 3-acetyltrichothecenes, 3-acetylnivalenol (3-ANIV), 3,15-diacetylnivalenol (3,15-diANIV), and 3,4,15-triacetylnivalenol (3,4,15-triANIV). When these trichothecenes were fed to FGD5, they were all metabolized to 4,15-diANIV and 4-ANIV, the final products of *F. graminearum* MAFF 111233 in liquid culture. Hexoside conjugates of NIV derivatives were not detected by LC-MS/MS analysis. In contrast, a very weak activity to conjugate glucose became detectable in the *Tri101* disruption mutant. FGD5/101 yielded an extremely limited amount of probable hexose conjugate(s) by each feeding, except for the 3,4,15-triANIV feeding, as described below.

In feedings with NIV and 3-ANIV, we found peaks of the parent ion *m*/*z* 492.2027 (Figure 6a; panel 1) and *m*/*z* 492.2019 (Figure 6a; panel 2) peaks, with their fragments at *m*/*z* 313.1250 and *m*/*z* 313.1276, respectively, of putative [NIV-hexoside (C_21_H_30_O_12_) + NH_4_]^+^ (theoretical *m*/*z* of 492.2076). Their MS/MS spectra were similar to the spectra of [NIV + NH_4_]^+^ (theoretical *m*/*z* of 330.1547) below an *m*/*z* value of 313.1278 (Figure 6a; panel 3). In feedings with 15-ANIV, 3-ANIV, and 3,15-diANIV, the parent ion peaks of *m*/*z* 534.2152 (Figure 6b; panel 1), *m*/*z* 534.2153 (Figure 6b; panel 2), and *m*/*z* 534.2147 (Figure 6b; panel 3) of putative [C_23_H_32_O_13_ (15-ANIV-hexoside) + NH_4_]^+^ (theoretical *m*/*z* of 534.2187) were detected. Their MS/MS spectra, including fragments *m*/*z* 337.1260, 337.1285, and 337.1270, respectively, were similar to those of 15-ANIV below an *m*/*z* value of 337.1279 (Figure 6b; panel 4). Due to the very poor efficiency of sugar conjugation, the structures of the putative NIV-hexoside and 15-ANIV-hexoside could not be confirmed by NMR. However, as the FGD5/101 mutant glucosylated C-4 of HT-2, it seems likely that these transformed products are NIV-4-*O*-glucoside and 15-ANIV-4-*O*-glucoside. If this is the case, t-type type B 3-acetyltrichothecenes, as well as 3-A HT-2, must be deacetylated to be conjugated with hexose because they are not good substrates of *Fusarium* UGT(s).

In a previous study, NIV-3-*O*-β-glucopyranoside (NIV-3-glc) was isolated from NIV-contaminated wheat and its structure was elucidated by NMR [40]. Thus, we applied the NIV-3-glc standard to LC-MS/MS and compared the spectrum of the ammonium adduct ion with that of the NIV-hexoside obtained in our feeding experiments (Appendix A). The MS/MS spectra were indistinguishable, probably because fragmentation of the parent ion predominantly occurs at the glycoside linkage between NIV and hexose, resulting in the formation of the same major fragments. However, the retention times were slightly different; NIV-hexoside obtained in our feeding assay was eluted at 5.70 min, which is earlier than the NIV-3-glc standard eluted at 5.95 min by the same HPLC system. The result demonstrates that the NIV-hexoside is not the known NIV-3-glc, which supports the above consideration of its structure.

### 2.5. Stability of the Glucoside Conjugates of Trichothecenes

Trichothecene 3-*O*-glucosides escape detection by the methods validated for trichothecene analysis [28]. Thus, the development of chemical and enzymatic hydrolysis of the modified mycotoxins has been attempted [41]. However, acidic and alkaline conditions were inefficient in cleaving the glucosidic bond of the modified mycotoxins. In addition, except for a specific glucosidase produced by a strain of *Bifidobacterium adolescentis*, no commercially available enzymes release trichothecene aglycon by hydrolysis [42]. With regard to the 4-*O*-glucosidic bond of trichothecenes, we also observed a similar stability of the glucoconjugates against a variety of enzymes obtained from Amano Enzyme Inc. (Nagoya, Japan). In this study, the chemical stabilities of **1**, **2**, and **3** values were studied at various pH.

LC–MS/MS analysis revealed differences in stability among the three trichothecene-4-*O*-glucosides. The 4-*O*-glucosidic bond of each compound was stable at every pH unit from pH 2 to 13. In contrast, the aglycons of **1** and **3** were not stable. Under acidic conditions for 3 d, a portion of **1** (retention time of 2.77 min) was converted to **4** (retention time of 2.34 min), most efficiently at pH 2. In MS/MS analysis, the parent ion *m*/*z* 448.2530 peak of **4**, corresponding to [C_21_H_34_O_9_ + NH_4_]^+^ (theoretical *m*/*z* 448.2541), was greater than that of [**1** (C_21_H_32_O_8_) + NH_4_]^+^ (theoretical *m*/*z* 430.2435) by *m*/*z* unit of 18.0106 (Figure 7; compare a and b). The MS peak of [TDmol + NH_4_]^+^ (theoretical *m*/*z* 268.1907) was not detected in the information-dependent acquisition (IDA) mode, demonstrating that the glycosidic linkage in **1** was not cleaved, but the aglycon of **1** was transformed into a new compound at pH 2. In contrast, **2**, a C-8 oxygenated derivative of **1**, was quite stable. These results suggest that the glycosidic linkage at C-4 of **1** and **2** was stable over a wide pH range but TDmol, aglycon of **1**, was rather unstable at a low pH. 

At pH 2, TDmol (C_15_H_22_O_3_) was converted into a new substance within 24 h; a theoretical fragment of *m/z* 269.1747 (detected as *m/z* 269.1750 in Figure 7c) derived from the parent ion of [C_15_H_24_O_4_ + NH_4_]^+^ (theoretical *m/z* 286.2013) was observed. Thus, the 4-*O*-glucosidic linkage of **1** stabilizes its trichothecene skeleton to some extent. The MS/MS spectrum of the acid-treated TDmol below an *m/z* value of 269.1750 (Figure 7c) was similar to that of **4** below an *m/z* value of 269.1736 (Figure 7b), suggesting that the aglycon of **1** was unstable at pH 2, irrespective of the presence or absence of the sugar moiety. Similar transformation was not observed for TCC, an aglycon of **2**.

To elucidate the structure of **4** by NMR, 4 mg of **1** was incubated at pH 2 for 21 d, when the transformation of **1** into **4** was completed. Two mg of highly purified **4** was obtained by purification with preparative HPLC. NMR analysis revealed the aglycon portion and overall structure of **4** to be 2,4,13-trihydroxyapotrichothecene (THA) [43] and THA-4-*O*-α-glucopyranoside (THA-4-glc) (**4**), respectively (Figure 8 and Appendix A, Appendix A). Thus, it was clarified that the trichothecene skeleton of **1** was opened and recyclized to yield an apotrichothecene skeleton of **4**.

With regard to **3**, we observed the parent ion *m*/*z* 562.2863 and *m*/*z* 478.2264 peaks of ammonium adduct ions of mono- and non-acylated products, corresponding to [T-2 triol-hexoside (C_26_H_40_O_12_) + NH_4_]^+^ (theoretical *m*/*z* 562.2858) and [T-2 tetraol-hexoside (C_21_H_32_O_11_) + NH_4_]^+^ (theoretical *m*/*z* 478.2283), respectively, at a pH greater than 9 (Appendix A). Comparison of their MS/MS spectra with those of their corresponding aglycons, [C_20_H_30_O_7_ (T-2 triol) + NH_4_]^+^ and [C_15_H_22_O_6_ (T-2 tetraol) + NH_4_]^+^, confirmed that deacylations, but not glycosidic linkage cleavage, occurred under the alkaline conditions.

In short, the trichothecene skeleton and side-chain ester linkages were sensitive to pH changes, as exemplified by the aglycons of **1** and **3**. However, cleavage of the C-4 glucosidic linkage does not proceed easily under wide pH ranges.

### 2.6. Cytotoxicity of the Glucoside Conjugates of Trichothecenes

In animal systems, glycoconjugates of toxic compounds produced by phase II reactions are generally less toxic than aglycones. In fact, trichothecene-3-glucosides are less toxic than their aglycones [28,44], although the risk of toxic aglycons being released must be considered. To date, nothing is known about the toxicity of trichothecene-4-glucosides.

By using HL-60, human myeloma leukemia cells, toxicities of the glucosides and aglycons were evaluated (Figure 9a). As expected, glucoconjugates **1**, **2**, and **3** showed marginal or much less growth inhibition, while the corresponding aglycones inhibited growth with IC_50_ of 4.04 ± 0.41, 9.00 ± 0.87, and 0.0183 ± 0.002 µM, respectively. No aglycones were detected in the cell culture medium. The toxicity of apotrichothecene glucoside **4** was marginal even at 10 µM.

We also performed a toxicity assay using a highly sensitive yeast bioassay for trichothecene detection using a *Saccharomyces cerevisiae* triple gene-deletion mutant [45]. As shown in Figure 9b, each glucoconjugate was much less toxic than its aglycone (IC_50_ values; TDmol, 3.46 ± 0.30; TCC, 5.01 ± 0.57; HT-2, 0.102 ± 0.020 µM), and all the glucoconjugates were nontoxic at 10 µM.

Interestingly, **3** was much less toxic than 3-A HT-2, which showed IC_50_ as low as 0.295 ± 0.26 µM and 3.45 ± 2.31 µM against HL-60 cells and the trichothecene-hypersensitive yeasts, respectively. The higher toxicity of 3-A HT-2 may be attributed to the instability of the 3-*O*-acetyl group as we previously reported [15].

### 2.7. Possible Role of Trichothecene 4-O-Glucosylation by Fusarium

The 3-*O*-acetylation of trichothecenes by Tri101p plays a critical role in the self-defense against trichothecene-producing *Fusarium*. In view of the activity of the *Tri101*-disruption strain, but not the wild-type strain, to conjugate glucose at C-4 in the liquid culture, the modification could be interpreted as *Fusarium* possessing a phase II xenobiotic metabolism of 4-*O*-glucosylation against C-3 unacylated trichothecenes. In mammals, phase II reactions involve conjugation with sulfate, glucuronide, glutathione, and glycine, but reports of glucoside conjugation are limited [46,47]. However, compared with the mammalian phase II reactions, glucoside conjugation is a well-known metabolic pathway for filamentous fungi. For example, *Cunninghamella elegans* can conjugate glucose to various polycyclic aromatic hydrocarbons after hydroxylation by phase I reactions [48]. Other fungi, including *Rhizopus* sp. [49], *Phanerochaete chrysosporium* [50], *Pleurotus ostreatus* [51], *Penicillium canescens* [52], *Mucor circinelloides* [53], *Phlebia radiata* [54], and *Mucor plumbeus* [55] are also known to have glucose-conjugation activity against xenobiotics.

The *Fusarium* phase II glucoconjugation system of trichothecenes demonstrated in this study is similar to the plant detoxification system against exogenously added trichothecenes [56]. However, the fungal and plant mechanisms differ; while the *Fusarium* system is based on the formation of α-glucosidic linkages mainly at C-4 of d-type trichothecenes oxygenated at C-4, the plant system targets t-type trichothecenes by means of 3-*O*-glucosylation. In addition, the stereochemistry of glucoconjugation varies among plants. While UGT forms glucoside conjugates with β-glucosidic linkage in rice and barley [27], the enzyme in oats forms glucoside conjugates of T-2 toxin with α-glucosidic linkage [57].

Although 3-acetyltrichothecenes were not conjugated with glucose at C-4 by fungal UGTs, the final trichothecene metabolites of the NIV chemotype cultured on solid substrate are 3-hydroxytrichothecenes that may be conjugated with sugars. Given that a small amount of NIV within the fungal cells undergoes glucoside conjugation under certain conditions, NIV-4-*O*-glucoside in cereal grains may also need to be monitored to ensure food safety.

## 3. Materials and Methods

### 3.1. Strains

Strain MAFF 111233 (NIV chemotype) and its gene disruptants, NBRC 113181 (*Fgtri11^−^* #2s1) [34], NBRC 113182 (*Fgtri7^−^*-#1s1*) [58], NBRC 113185 (*Fgtri101*^−^ #2S1*) [59], and NBRC 114123 (FGD8) [36], are previously described strains deposited at the Biological Resource Center, NITE (NBRC) (Chiba, Japan), which were used for the preparation of trichothecene substrates necessary for the feeding experiments. Strain MAFF 101551 (3-ADON chemotype) and its transgenic strain NBRC 113183 (P*_TEF1α_*::FgTri13__3-ADON chemotype_-*neo* [*Fgtri13_*o/e #5]) [36], *F. sporotrichioides* NBRC 9955 (T-2 toxin producer) [60], *Spicellum roseum* JCM 8964 (8-deTCN producer) [61], and *Trichothecium roseum* NBRC 31647 (TCN producer) were also used for the preparation of trichothecene substrates. *Bacillus* spp. LC466619 was used for the specific deacetylation of T-2 toxin to obtain HT-2. *Escherichia coli* (BL21 [DE3]-pColdIII-Tri101) was used to obtain recombinant Tri101p [62]. The *F. sporotrichioides* strain FsΔ*Tri5* (Δ*Fstri5::hph*), derived from NBRC 9955 [35], and *F. graminearum* strains NBRC 113175 (Δ*Fgtri5::neo*; FGD5) [6] and NBRC 114124 (FGD5/101) [36], derived from MAFF 111233, were used for the feeding experiments.

### 3.2. Reagents

Hygromycin B and geneticin disulfate used for the maintenance of fungal disruptants were obtained from Nacalai Tesque, Inc. (Kyoto, Japan). TLC plates (Silica gel 60 F_254_) and LC–MS grade acetonitrile were purchased from Merck KGaA (Darmstadt, Germany). For reagents for TLC analysis, hexane, toluene, NBP, and tetraethylenepentaamine (TEPA) were purchased from Kanto Chemical Co., Inc. (Tokyo, Japan).

### 3.3. Medium and Culture Conditions

Fungal strains were maintained on V8 juice agar (20% Campbell’s V8 juice [*v/v*], 0.3% CaCO_3_ [*w/v*], and 2% agar [*w/v*]) containing appropriate antibiotics, if necessary. For trichothecene production, hyphae were transferred to an appropriate liquid medium. YS_60 medium (0.1% yeast extract [*w/v*] and 6% sucrose [*w/v*]) [36,58] was used for JCM 8964, NBRC 31647, and MAFF 101551 to produce trichothecenes. RF medium [63] was used for MAFF 111233 and its transformants for production of NIV-type trichothecenes and trichothecene intermediates. GYEP medium (0.1% yeast extract [*w/v*], 5% glucose [*w/v*], and 0.1% peptone [*w/v*]) was used for T-2 toxin production by NBRC 9955.

### 3.4. Preparation of Trichothecene Substrates for the Feeding Experiment

To prepare 8-deTCN and TCN, strains JCM 8964 and NBRC 31647, respectively, were cultured on YS_60 medium with gyratory shaking at room temperature for 7 d. TDmol and TCC were prepared by treating 8-deTCN and TCN with 2.8% ammonium solution, respectively, at 37 °C for 2 d.

For production of NIV-type trichothecenes (except for 3-ANIV) and the intermediates, the hyphae of *F. graminearum* MAFF 111233 and its gene disruption mutants were inoculated into RF medium. They were incubated with gyratory shaking at 20–22 °C for the appropriate incubation periods [34]. The wild-type strain of MAFF 111233 was incubated in RF medium for 3–7 d and 7–10 d, to obtain the maximum amount of 4,15-diANIV and 4-ANIV, respectively. NIV was produced by treating 4-ANIV with 0.5% ammonium solution at 37 °C for 1 h. ITDmol was prepared from the NBRC 113185 culture after 7-day incubation, while ITD was produced by NBRC 113181 after 7-day incubation [15,35]. NBRC 113182 produced 3,15-diANIV after 3–6-day incubation and 15-ANIV after 21-day incubation. NBRC 114123 produced 3,4,15-triANIV after 5-day incubation.

The hyphae of NBRC 113183 were inoculated into YS_60 medium and incubated at room temperature with continuous shaking for 7 d, yielding mixtures of 3-ANIV and 3-ADON.

The conidia of NBRC 9955 were inoculated into the GYEP medium and incubated at 28 °C for 5 d to produce T-2 toxin. HT-2 was obtained by incubating T-2 toxin with the crude cell extract of *Bacillus* spp. LC466619, containing the deacetylase specific to C-4. For production of 3-A HT-2, rTri101p was used to acetylate T-2 toxin and HT-2, respectively, as previously described [34]: the reaction mixture contained approximately 2.0 µM rTri101p, 100 µmol acetyl CoA trilithium salt, and 10 mg each of T-2 toxin (21.5 µmol) or HT-2 (23.6 µmol) in 20 mL of 50 mM Tris-HCl buffer (pH 7.5). The reaction was performed at 30 °C for 3 h.

### 3.5. Extraction and Purification of Trichothecenes

The transformation of each toxin in the feeding experiment was confirmed using TLC. An aliquot of each fungal culture was extracted with an equal volume of ethyl acetate. For the extraction of hydrophilic sugar conjugates, an equal volume of acetonitrile was added, followed by the addition of a small amount of 5 M NaCl aqueous solution to separate the organic phase. The extract was dried, dissolved in ethanol, and subjected to TLC. The TLC plates were developed with ethyl acetate/toluene (3:1) or ethyl acetate/toluene/hexane (2:2:1). The toxins were visualized using the NBP/TEPA method [64,65].

For the large-scale extraction of hydrophobic trichothecenes, an equal volume of ethyl acetate was added to the culture and the extraction was repeated. With regard to the extraction of the less hydrophobic NIV transformed from 4-ANIV, an equal volume of acetonitrile was added to the solution following the evaporation of the ammonium in the reaction mixture by the continuous flow of N_2_. A small amount of 5 M NaCl (aq) was then added to separate the solutions into two phases.

The organic phase was concentrated using a rotary evaporator or Smart Evaporator (BioChromato, Inc., Kanagawa, Japan) and the condensate was dissolved in ethyl acetate and applied to Purif-Rp2 equipped with Purif-Pack SI 25 (Shoko Scientific, Kanagawa, Japan). A portion of each fraction was applied on TLC, and the fractions containing each trichothecene were collected and concentrated again. The condensate was dissolved in ethanol and filtered through a 0.22 µm polytetrafluoroethylene (PTFE) syringe filter (ANPEL Laboratory Technologies Inc., Shanghai, China). The filtrate was applied to preparative HPLC (LC-4000 series, JASCO, Corp., Tokyo, Japan; UV detection at 254 nm or 195 nm) equipped with a C_18_ column (Pegasil ODS SP100 10φ × 250 mm; Senshu Scientific Co., Ltd., Tokyo, Japan). Trichothecenes were eluted in a mobile phase of H_2_O-acetonitrile using the optimized gradient mode for individual toxins at 40 °C at a flow rate of 3 mL/min. The purification step using preparative HPLC was repeated until the purity of the target compound was >98%.

### 3.6. HPLC and LC-MS/MS Analysis

Each purified trichothecene was applied to analytical HPLC (LC-2000 plus series; JASCO, Corp., Tokyo, Japan; UV detection at 254 nm or 195 nm) equipped with a C_18_ column (Pegasil ODS SP100 4.6φ × 250 mm). Trichothecenes were eluted in a mobile phase of H_2_O-acetonitrile using an appropriate gradient mode at 40 °C at a flow rate of 1 mL/min. The concentration of each trichothecene was calculated from the corresponding peak area based on the standard curve, if available. If the standard curve was not available, each purified toxin was dried completely using a freeze dryer (TITEC Corp., Saitama, Japan) and weighted. In some cases, quantitative NMR was performed, as previously described [66].

For LC–MS/MS analysis, samples were separated using an Eksigent ekspert™ ultraLC 100-XL system (Dublin, CA, USA) with a C_18_ reverse phase column (PEGASIL ODS SP100-3; 2φ × 100 mm) according to the following steps: a linear gradient of 10–95% of acetonitrile in 5 min in 0.1% ammonium formate at a flow rate of 0.3 mL/min. The ultraLC was connected to an AB SCIEX Triple TOF 4600 system (Framingham, MA, USA), with a DuoSpray source operated in electrospray ionization mode. The IDA and/or time-of-flight (TOF)–MS methods were used to obtain MS/MS spectra in the positive ion mode. Data were analyzed using the PeakView software version 1.2.0.3 (AB SCIEX).

### 3.7. Feeding Experiments and Purification of Trichothecene Metabolites

For feeding experiments of the d-type trichothecenes, conidia of FsΔ*Tri5* were pre-incubated in GYEP medium at 28 °C with reciprocal shaking; mycelial plugs of FGD5 were incubated in RF medium at 20–22 °C with gyratory shaking. After 2–4 d of incubation, the freshly prepared hyphae were washed in water, and the FsΔ*Tri5* and FGD5 hyphae were added to 50 mL of GYEP and RF media, respectively. One milligram of each d-type trichothecene (TDmol, TCC, 8-deTCN, and TCN in 100 µL ethanol) was immediately added and incubated with shaking. In parallel, the vehicle (ethanol) and 1 mg ITDmol in 100 µL ethanol were fed as the negative and positive controls, respectively. Two aliquots of 2 mL medium were taken at 0, 3, 6, 24, and 48 h; one aliquot was extracted with ethyl acetate for TLC analysis, and another was extracted with acetonitrile and NaCl (aq) for LC–MS/MS analysis. The developing solvents for TLC of metabolites of FsΔ*Tri5* and FGD5 were ethyl acetate:toluene:hexane (2:2:1) and ethyl acetate:toluene (3:1), respectively.

For feeding experiments of the t-type type A trichothecenes, the hyphae of *F. graminearum* FGD5 and FGD5/101 were prepared as described above, and 1 mg each of HT-2 and 3-A HT-2, respectively, was fed to 50 mL of RF medium containing freshly prepared hyphae in each flask. The strains FGD5 and FGD5/101, used for the feeding experiments, were confirmed to be metabolically active. FGD5 rapidly transformed ITDmol into NIV-type trichothecenes [34], and FGD5/101 similarly transformed TDmol into TDmol-4-glc [33]. The time course experiment was performed in a manner similar to that of the d-type trichothecene feedings. The extracts were analyzed by TLC and LC–MS.

For feeding experiments of the t-type type B trichothecenes, the hyphae were prepared in the same way as described above. The substrates fed to the hyphae were as follows: NIV, 3-ANIV, 15-ANIV, 3,15-diANIV, and 3,4,15-triANIV. The hyphae were incubated with 1 mg of each substrate in 50 mL of RF medium and each culture medium was sampled on days 0, 2, 5, 7, and 14. The extracts were applied to TLC and LC–MS.

To prepare sufficient amounts of trichothecene glucoside conjugates for NMR analysis, we performed large-scale feeding experiments. For the production of **1** and **2**, 8 mg each of TDmol and TCC was added to 200 mL of GYEP media in 500 mL Erlenmeyer flasks, respectively, and freshly prepared FsΔ*Tri5* hyphae were inoculated onto the media. The hyphae were incubated at 28 °C for 2 d with vigorous shaking. A total of 2 L and 1.8 L culture medium were collected for purification **1** and **2**, respectively. For the production of **3**, 4 mg of HT-2 was added to 200 mL of RF media in 500 mL flasks and freshly prepared FGD5/101 hyphae were inoculated. The hyphae were incubated at 20–22 °C for 16 d with vigorous shaking. A total of 15 L of culture medium was collected for the production of **3**.

For the extraction and purification of the glucoside conjugates **1**, **2**, and **3**, acetonitrile equal in volume to that of each culture medium was added, filtered through filter paper (Toyo Roshi Kaisha, LTD., Tokyo, Japan), and mixed with a small amount of 5 M NaCl aq to separate into two phases. The aqueous phase was extracted again with acetonitrile, and the organic phase was collected and concentrated. The condensate of each glucoside conjugate was dissolved in ethanol and applied to Purif-Rp2 equipped with Purif^®^-Pack ODS-25 size 60. The trichothecene metabolites were eluted with a mobile phase of acetonitrile and water in an appropriate gradient mode and the fractions containing each target conjugate were concentrated.

The condensate was dissolved in ethanol and applied to a preparative HPLC (UV detection at 195 nm). This purification step was repeated three times for **1** and **2**, and five times for **3**, following the procedure described in Section 3.5. Each purified conjugate (**1**, **2**, and **3**) was dried completely using a freeze dryer (TITEC Corp., Saitama, Japan), weighed, and subjected to NMR spectroscopy.

### 3.8. In Vitro Trichothecene 3-O-Acetyltransferase (Tri101p) Activity Assay

To evaluate the 3-*O*-acetylation activity of Trip101p against HT-2 and its glucoconjugate, 20 µg of HT-2 and 27.6 µg of HT-2-4-glc (each being 47.1 nmol) were incubated with acetyl CoA (200 nmol) and rTri101p (0.2 and 2.0 µM) in 10 mM Tris-HCl buffer (pH 7.5) at 30 °C for 1 and 6 h. rTri101p inactivated in advance by boiling was used as a negative control. The product was extracted with acetonitrile by the addition of 5 M NaCl aq, dissolved in ethanol, and filtered through a 0.22 µm PTFE syringe filter. Each filtrate was subjected to LC-MS/MS analysis.

### 3.9. Stability of the Glucoside Conjugates of Trichothecenes and Trichothecene Aglycons

The pH of each solution was adjusted to pH 2 to pH 13 at every pH unit by adding HCl or ammonium solution, and 5 µg of each glucoside conjugate (**1**, **2**, and **3**) was added to 300 µL of each solution. After incubation at 37 °C for 3 d, the reaction mixture was diluted 10 times with ethanol. Two microliters of each sample was applied to LC-MS/MS, and data were acquired in IDA mode. To investigate the acid stability of the aglycon TDmol, we also added 5 µg of each trichothecene (TDmol and TCC) into 300 µL of solution (pH 2), and incubated at 37 °C for 3 d. The solution was diluted 10 times with ethanol, and 2 µL each was applied to LC-MS/MS. The data were acquired in the TOF/MS mode.

To prepare a sufficient amount of apotrichothecene glucoside conjugate (**4**) for NMR analysis, we added 4 mg of **1** to 16 mL of acetic acid solution (pH 2.0) and incubated it at 37 °C for 3 weeks. Under the condition, all of the compound **1** disappeared and transformed into compound **4**. The reactant was concentrated using a rotary evaporator, dissolved in 50% ethanol, and filtered through a 0.22 µm PTFE syringe filter. The filtrate was subjected to a preparative HPLC, as described in Section 3.5. Purified compound **4** was dried completely using a freeze dryer, weighed, and subjected to NMR spectroscopy.

### 3.10. NMR

All NMR spectra were recorded on a JEOL JNM-ECX500 (500 MHz) spectrometer in deuterated methanol (MeOH-d_3_) calibrated with a solvent peak at 3.31 ppm for the measurement of H nucleus and 49.3 ppm for that of C nucleus. Purified trichothecenes conjugates were identified by some analyses of ^1^H NMR, ^13^C NMR, correlation spectroscopy (COSY), heteronuclear multiple quantum coherence (HMQC), and heteronuclear multiple bond coherence (HMBC). If necessary, total correlation spectroscopy (TOCSY) or 1D-nuclear overhauser effect (1D-NOE) analysis was performed.

### 3.11. Cytotoxicity Assay

HL-60 cells were grown in RPMI1640 medium (Nacalai Tesque Co., Inc., Kyoto, Japan) containing 20% fetal bovine serum (FBS; Biowest, Nuaillé, France), antibiotics (100 U/mL for penicillin and 100 µg/mL streptomycin), 1 mM sodium pyruvate, and 100 µM 2-mercaptoethanol. In addition to the normal heat inactivation of FBS at 56 °C for 30 min, it was briefly boiled to inactivate the contaminating deacetylase, as described previously [15].

For the cytotoxic assay, cells in the logarithmic growth phase were prepared. To each well of a 96-well microtiter plate, 95 µL of culture cells (2.5 × 10^4^/well) were seeded. Five microliters of each test compound in 10% ethanol (vehicle) was added and the cells were incubated for 2 d in a CO_2_ incubator. The final concentrations of each trichothecene were as follows: TDmol (0.13–25 µM), TDmol-4-glc (**1**) (1.4–351 µM), TCC (0.14–27.8 µM), TCC-4-glc (**2**) (0.60–150 µM), HT-2 (0.0067–0.134 µM), HT-2-4-glc (**3**) (0.19–46.8 µM), 3-A HT-2 (0.04–10.9 µM), and THA-4-glc (0.57–142 µM). The in vitro cytotoxicity of each test compound was measured using a 2-(2-methoxy-4-nitrophenyl)-3-(4-nitrophenyl)-5-(2,4-disulfophenyl)-2H-tetrazolium (WST-8) assay using a CCK-8 kit (Dojindo Laboratories, Kumamoto, Japan). After incubation with each trichothecene for 2 d, 10 μL of WST-8 solution was added to 100 μL cell cultures and incubated at 37 °C for 3.5 h in a CO_2_ incubator. Absorbance (A_450_) was measured using a Multiskan FC microplate reader (Thermo Fisher Scientific, Waltham, MA, USA). The percentage of inhibition of growth was calculated as:$$\frac{A450\;\left(\mathrm{vehicle}\right)-A450\;\left(\mathrm{test}\;\mathrm{compound}\right)}{A450\;\left(\mathrm{vehicle}\right)-A450\;\left(\mathrm{blank}\right)}\times 100$$

These experiments were performed in triplicate.

For a highly sensitive yeast bioassay for trichothecenes, the triple null mutant strain, *pdr5*Δ *erg6*Δ *rpb4*Δ of *S. cerevisiae* BY4742 (MATα *his3*Δ*1 leu2*Δ*0 lys2*Δ*0 ura3*Δ*0*, Horizon Discovery Ltd., Cambridge, UK) was preincubated in 5 mL of YPD medium (1% yeast extract, 2% peptone, and 2% glucose) for 18–48 h with shaking at 30 °C. The OD_620_ of the cell culture was adjusted to 0.1 in YPD medium containing 0.003% sodium dodecyl sulfate (SDS) and 196 µL of cell culture was transferred to each well of the microtiter plate. Thereafter, 4 µL of inhibitor solution (10% ethanol as a vehicle) was added to each well and the yeast cells were incubated at 30 °C for 24 h with continuous shaking. The final concentrations of each trichothecene in each well were as follows: TDmol (0.05–25 µM), TDmol-4-glc (0.24–119 µM), TCC (0.056–27.8 µM), TCC-4-glc (0.24–120 µM), HT-2 (0.0027–1.07 µM), HT-2-4-glc (0.075–37.4 µM), 3-A HT-2 (0.017–43.7 µM), and THA-4-glc (0.23–113 µM). The OD_620_ for trichothecene-treated and vehicle-treated cultures was measured using a Multiskan FC Microplate Photometer and the value was subtracted from the corresponding OD_620_ at a time point of zero. The relative growth rate was calculated as the OD_620_ ratio of the trichothecene-treated samples to the untreated controls. The percentage of inhibition of growth was calculated as:$$100\;-\;\frac{\Delta A620\;\left(\mathrm{test}\;\mathrm{compound}\right)}{\Delta A620\;\left(\mathrm{vehicle}\right)}\;\times 100$$
where ΔA_620_ is the average change in absorbance of each well (end point A_620_ and start point A_620_). These experiments were performed in duplicate.

### 3.12. Statistical Analysis

The log-logistic model was applied using R version 3.5.0 (R project for statistical computing) for the calculation and statistical analysis of IC_50_ values.

## 4. Conclusions

The trichothecene-producing *Fusarium* species proved to have a fungal phase II reaction against exogenously added trichothecenes. Although the 4-*O*-glucosylation was limited in comparison to the major resistance mechanism of 3-*O*-acetylation, the modification significantly reduced the toxicity compared to the corresponding aglycons. Therefore, 4-*O*-glucosylation may serve as a phase II xenobiotic metabolism against 4-hydroxytrichothecenes for t-type trichothecene producers.

## Figures and Tables

**Figure 1 ijms-22-13542-f001:**
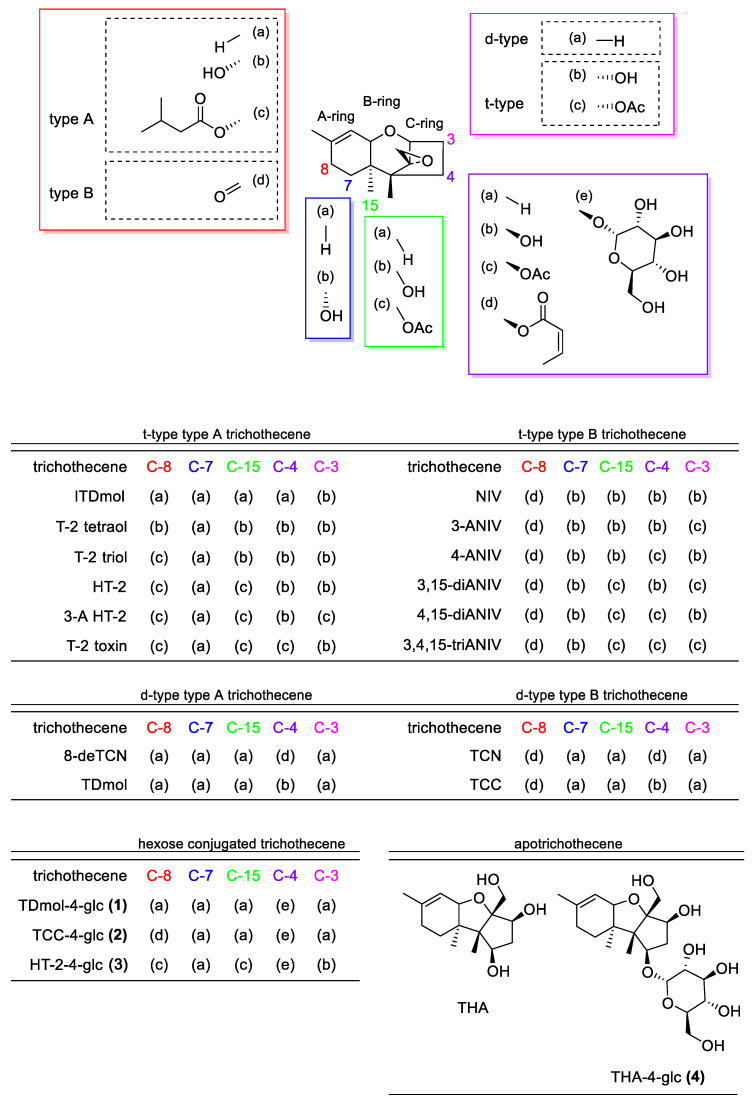
Chemical structures of trichothecenes and apotrichothecenes.

**Figure 2 ijms-22-13542-f002:**
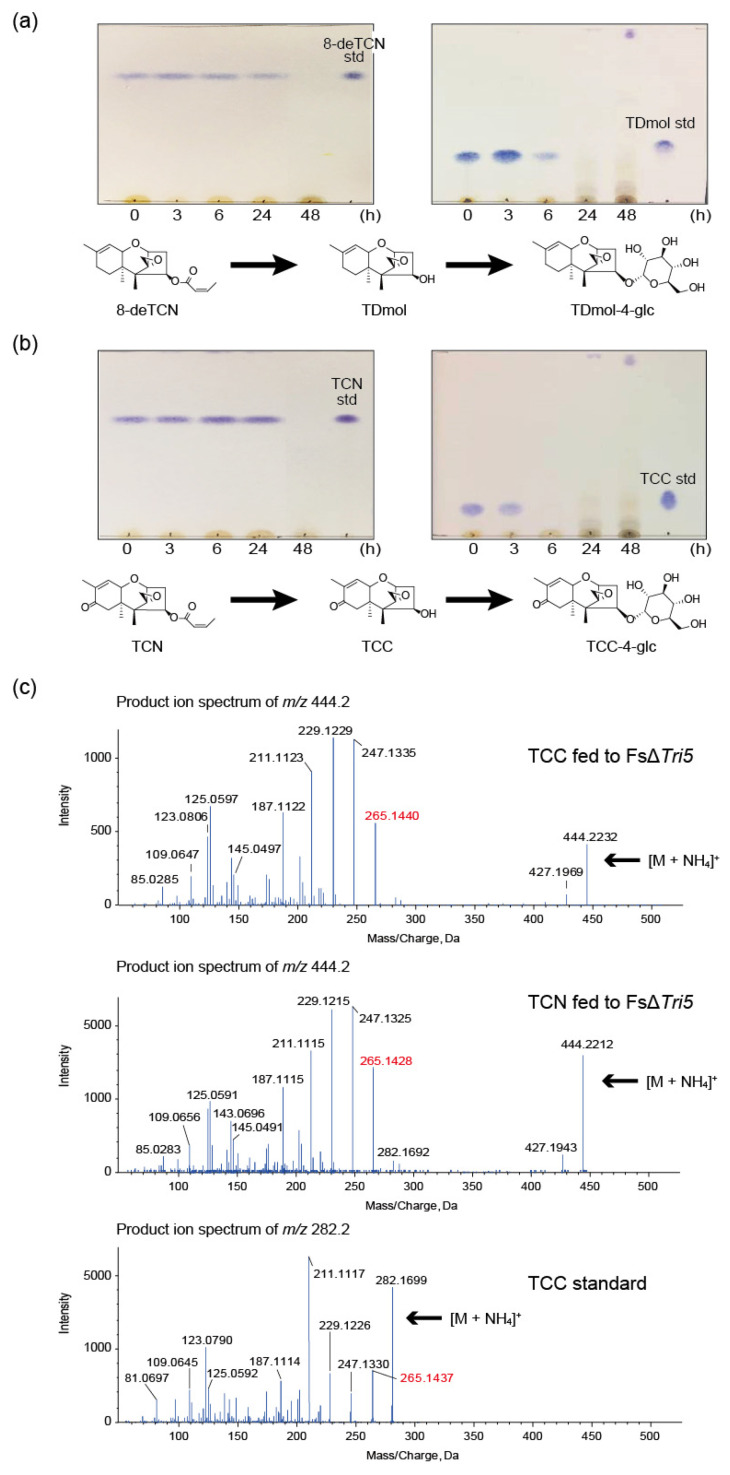
Hexose conjugation of d-type trichothecenes by a t-type trichothecene-producer, strain FsΔ*Tri5*. (**a**) Time course of the disappearance of 8-deoxytrichothecin (8-deTCN) (left) and trichodermol (TDmol) (right) spots on TLC. While 8-deTCN was fully recovered from the culture at 24 h after the feeding, TDmol significantly decreased by 6 h and was barely detectable at 24 h. A hypothetical scheme of transformation is depicted below the TLC. (**b**) Time course of the disappearance of the trichothecin (TCN) (left) and trichothecolone (TCC) (right) spots on TLC. While nearly full recovery of TCN was made at 24 h, TCC was barely recovered at 6 h. A hypothetical scheme of transformation is depicted below the TLC. (**c**) LC–MS/MS analysis of the acetonitrile extract of the culture fed with TCC and TCN. Aliquot of each culture supernatant at 48 h was extracted with acetonitrile and 5 M NaCl (Section 3.5). The dried and reconstituted sample was analyzed by LC-MS/MS. An MS peak corresponding to the theoretical *m*/*z* value of 444.2228 was detected at around retention time of 2.4 min in both the TCC (upper panel) and the TCN (middle panel) feedings. For comparison, the spectra of the ammonium adduct ion of TCC standard in LC-MS/MS (lower panel) is also shown.

**Figure 3 ijms-22-13542-f003:**
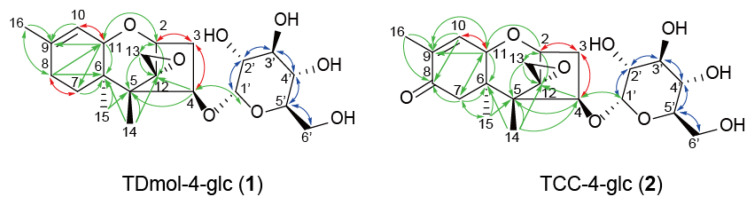
Determination of the structures of d-type trichothecenes **1** and **2**. The red, blue, and green arrows indicate ^1^H–^1^H correlation spectroscopy (COSY), total correlation spectroscopy (TOCSY), and heteronuclear multiple bond coherence (HMBC) correlations, respectively.

**Figure 4 ijms-22-13542-f004:**
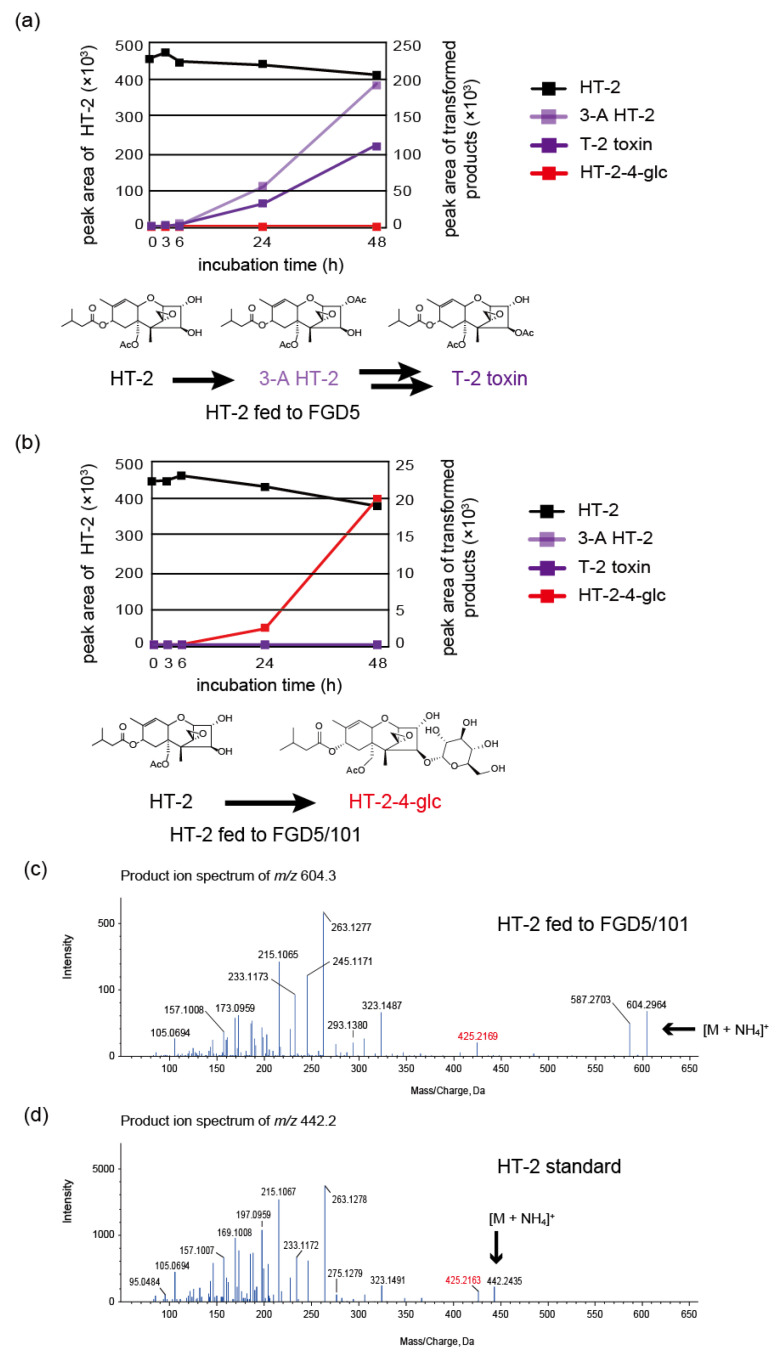
Metabolism of HT-2 toxin (HT-2) fed to the *Fusarium. graminearum* mutants, with or without functional *Tri101*. The fresh mycelia of FGD5 and FGD5/101 in rice flour (RF) liquid medium were fed with 20 µg/mL of HT-2 and cultured with gyratory shaking at 125 rpm and 25 °C. The culture was sampled at the indicated time and aliquots were used to monitor the changes in quantity. The metabolites of HT-2 were analyzed by the extracted ion chromatogram (XIC) mode in LC-MS/MS. (**a**) Time course changes of HT-2 and its transformed products fed to the FGD5 mutant. The values at the left and right perpendicular axes represent the peak area of the substrate HT-2 and possible products to be assessed [T-2 toxin, 3-acetyl HT-2 toxin (3-A HT-2), and HT-2 toxin-4-*O*-α-glucopyranoside (HT-2-4-glc)], respectively. (**b**) Time course changes of HT-2 and its transformed products fed to the FGD5/101 mutant. The values at the left and right perpendicular axes represent the peak area of the substrate HT-2 and possible products to be assessed (T-2 toxin, 3-A HT-2, and HT-2-4-glc), respectively. (**c**) The MS/MS spectrum of [HT-2-4-glc + NH_4_]^+^ detected in the feeding sample of the FGD5/101 mutant at 48 h. (**d**) The spectrum of [HT-2 + NH_4_]^+^ in our in-house MS/MS library is shown for comparison.

**Figure 5 ijms-22-13542-f005:**
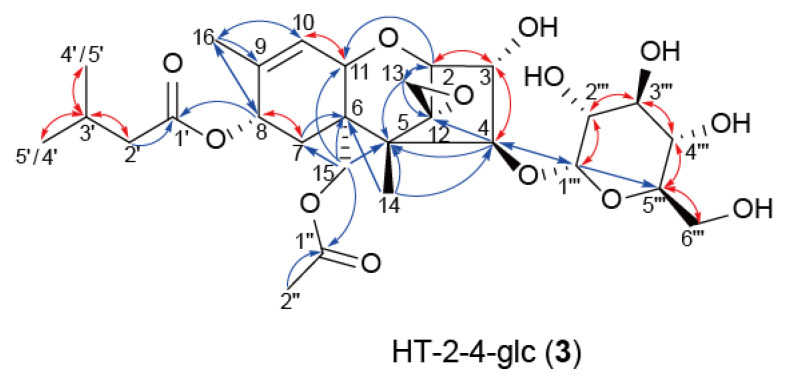
Determination of the structure of **3**. The red and blue arrows indicate ^1^H–^1^H correlation spectroscopy (COSY) and heteronuclear multiple bond coherence (HMBC) correlations, respectively.

**Figure 6 ijms-22-13542-f006:**
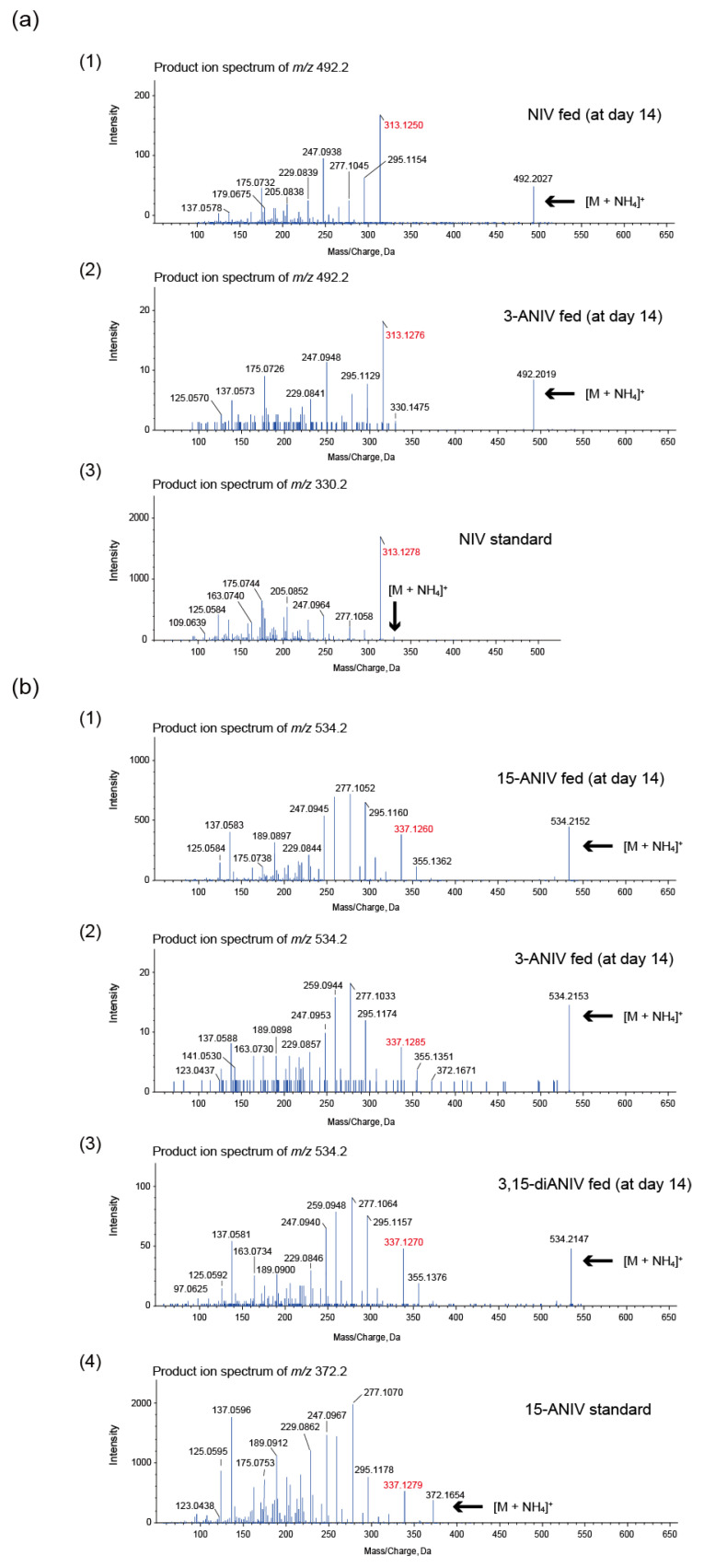
LC–MS/MS analysis of the transformed products of nivalenol (NIV)-type trichothecenes by FGD5/101. The transformed trichothecenes were analyzed 14 d after the feedings. The ammonium adduct ion of each transformed metabolite is shown. (**a**) MS/MS analysis of hexose-conjugated NIV obtained by feedings of NIV (panel 1) and 3-acetylnivalenol (3-ANIV) (panel 2). The spectrum of [NIV (C_15_H_20_O_7_) + NH_4_]^+^ (theoretical *m*/*z* of 330.1547) (panel 3) is shown as a reference for comparison. (**b**) MS/MS analysis of hexose-conjugated 15-ANIV obtained by feedings of 15-acetylnivalenol (15-ANIV) (panel 1), 3-ANIV (panel 2), and 3,15-diacetylnivalenol (3,15-diANIV) (panel 3). The spectrum of [15-ANIV (C_17_H_22_O_8_) + NH_4_]^+^ (theoretical *m*/*z* of 372.1653) (panel 4) is shown as a reference for comparison.

**Figure 7 ijms-22-13542-f007:**
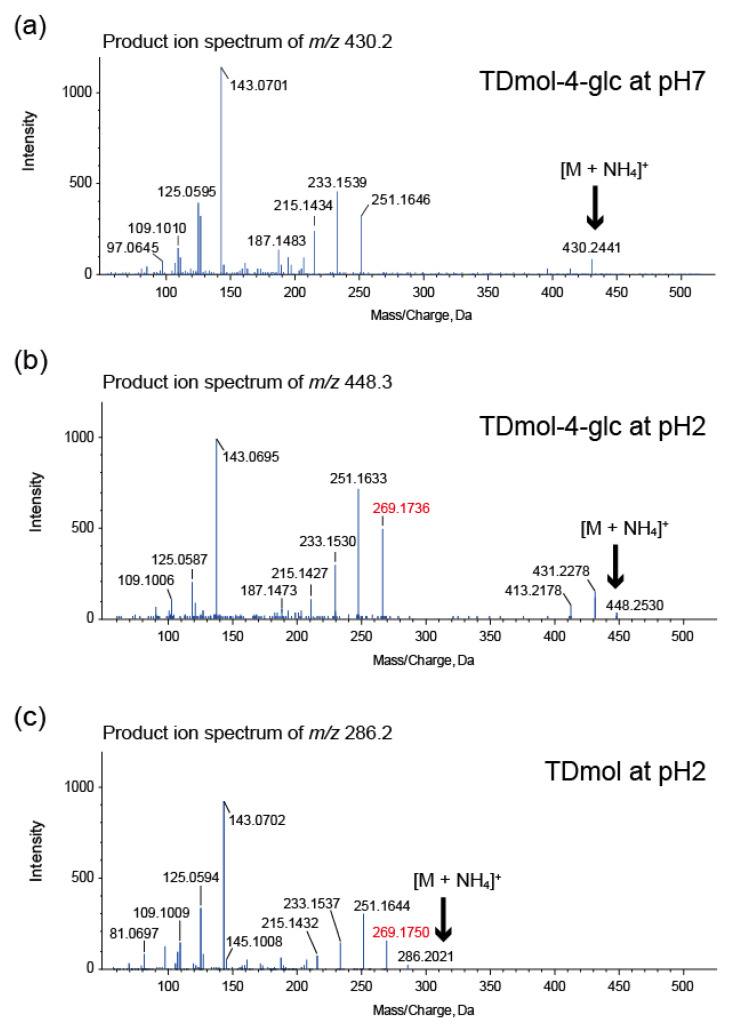
pH stability test of **1**. (**a**) MS/MS spectrum of the ammonium adduct ion of **1** incubated for 3 d at pH 7. (**b**) MS/MS spectrum of the ammonium adduct ion of **4**, which was partially formed by incubating **1** for 3 d at pH 2. (**c**) MS/MS spectrum of the ammonium adduct ion of trichodermol (TDmol) incubated for 3 d at pH 2.

**Figure 8 ijms-22-13542-f008:**
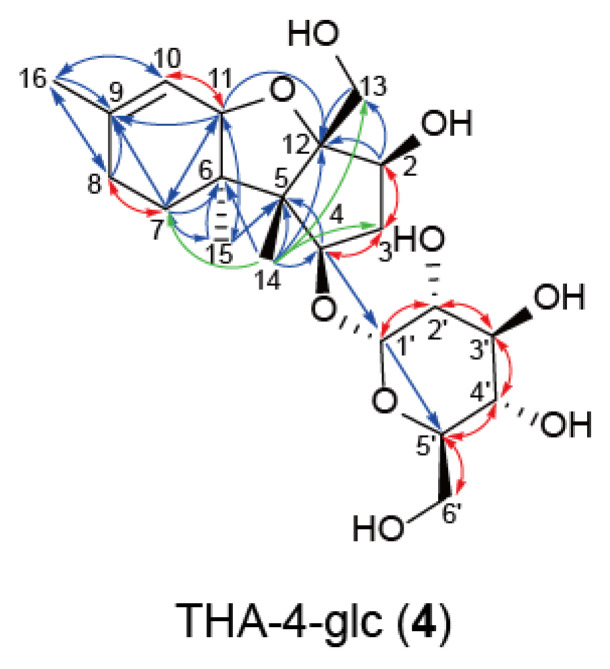
Determination of the structure of **4**. A C-8 unoxygenated d-type trichothecene glucoside conjugate, **1**, was completely converted to an apotrichothecene glucoside conjugate, **4**, by incubating for 21 d at pH 2. The red, blue, and green arrows indicate ^1^H–^1^H correlation spectroscopy (COSY), heteronuclear multiple bond coherence (HMBC), and nuclear overhauser effect (NOE) correlations, respectively.

**Figure 9 ijms-22-13542-f009:**
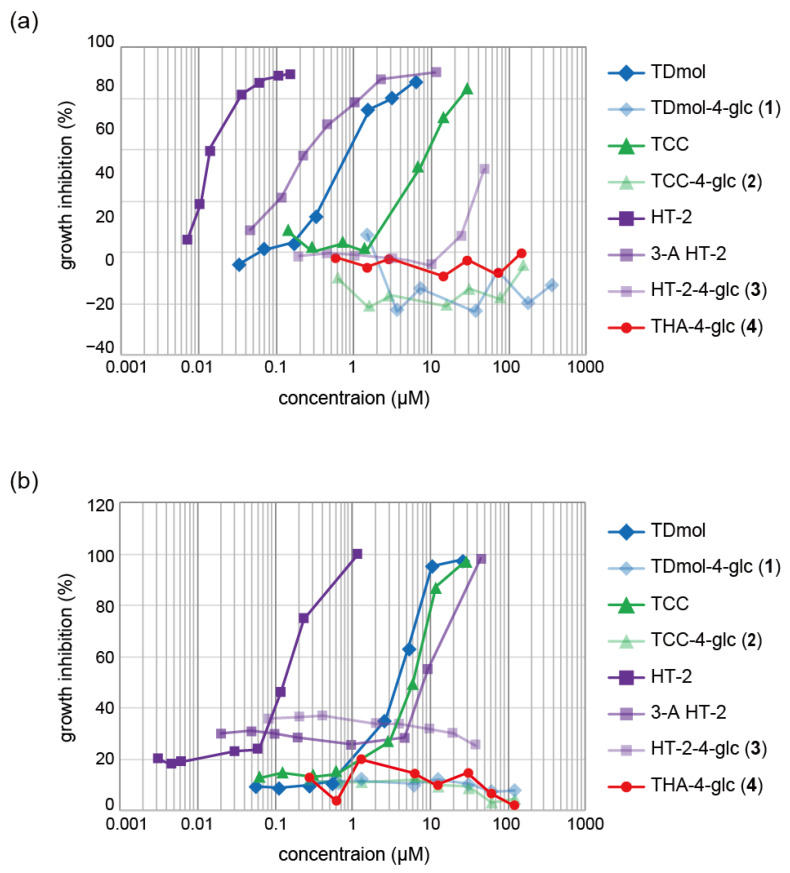
Dose-response cytotoxicity curves of trichothecenes. (**a**) Cytotoxicity assay using HL-60 cells. (**b**) Cytotoxicity assay using trichothecene hypersensitive yeast, *pdr5*Δ *erg6*Δ *rpb4*Δ of *Saccharomyces cerevisiae* BY4742 mutant. The assays of HL-60 and yeast cells were carried out in 96-well plates.

## Data Availability

Not applicable.

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
