# Peer review of "4-*O*-Glucosylation of Trichothecenes by *Fusarium* Species: A Phase II Xenobiotic Metabolism for t-Type Trichothecene Producers"

_ijms, 2021, doi:10.3390/ijms222413542_

Round 1
Reviewer 1 Report
The manuscript studies 4-O glycosylation as a phase II xenobiotic metabolism in the t-type trichothecene producers against d-type trichothecenes. The biological classification of trichothecene in t -type or d-type, based on the presence (t-type)or absence (d-type) of C-3 oxygen function was used. The data, conclusion, discussion are scientific sound and comprehensive. I suggest that this manuscript can be published in Toxins
Author Response
Thank you very much for your invitation to Toxins, however, this time, we would like to submit our manuscript for International Journal of Molecular Sciences. We have prepared it specifically for the special issue of “Molecular Biology and Chemistry of Mycotoxins and Phytotoxins”. Thus, we believe it to be perfectly suitable for our manuscript. Please understand our intention. We are under serious consideration to submit our next manuscript to Toxins.
Reviewer 2 Report
Introduction is too long and needs some corrections.
In Material and methods: Strains chapter needs some information how strains were identified.
Please specify the name as Bacillus sp. must be Bacillus spp. line 390, 431, 355
In line 336 Saccharomyces cerevisiae is with a different font size.
Tables and figures must be in text not after references.
Conclusion is not number 2.8, but 55. and must be at the end of manuscript after material and methods. Conclusion is very general and in conclusion there are no necessary references.
The article is very well written but the authors have to process the text according to the instructions for the authors.
Author Response
Thank you very much for the kind and helpful suggestions of the referee to improve our manuscript. We greatly appreciate it. Here, we answered to these suggestions one by one. We hope that the referee will accept our revised manuscript.
Introduction is too long and needs some corrections.
--->We shortened it according to the referee’s suggestion.
In Material and methods: Strains chapter needs some information how strains were identified.
Please specify the name as Bacillus sp. must be Bacillus spp. line 390, 431, 355
--->We corrected them.
In line 336 Saccharomyces cerevisiae is with a different font size.
--->We corrected it. There are some other parts with different font size, so we corrected them, too. We believe that mistakes happened after our submission.
Tables and figures must be in text not after references.
--->We corrected them.
Conclusion is not number 2.8, but 55. and must be at the end of manuscript after material and methods. Conclusion is very general and in conclusion there are no necessary references.
--->We corrected them.
The article is very well written but the authors have to process the text according to the instructions for the authors.
--->We corrected them.